# Equilibrated Diffusion: Frequency-aware Textual Embedding for Equilibrated Image Customization

Liyuan Ma
Westlake University
Hangzhou, China
maliyuan@westlake.edu.cn

Xueji Fang
Zhejiang University
Westlake University
Hangzhou, China
fangxueji@zju.edu.cn

Guo-Jun Qi*
Westlake University
Hangzhou, China
guojunq@gmail.com

Concept | Unstylized Prompt | Stylized Prompt

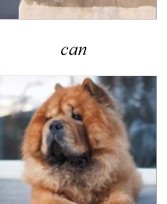
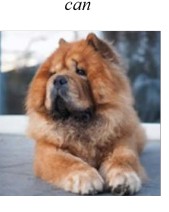
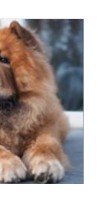

*can*

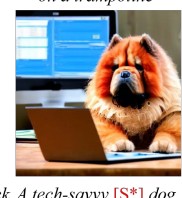
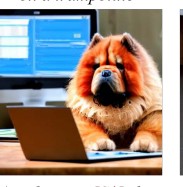

A [S*] can with the Eiffel Tower in the background

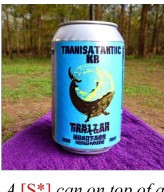
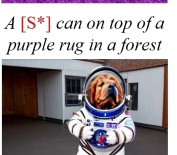
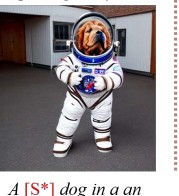

A [S*] can bouncing on a trampoline

A [S*] can on top of a purple rug in a forest

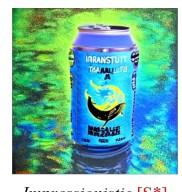
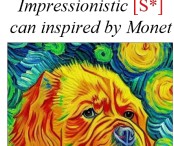

Impressionistic [S*] can inspired by Monet

Sketch of a [S*] can

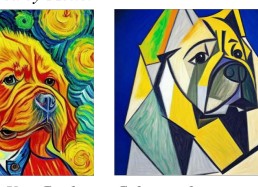
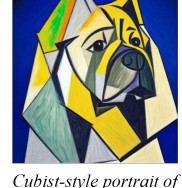

A red [S*] can

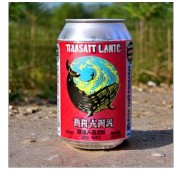
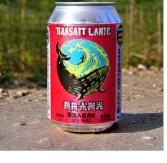
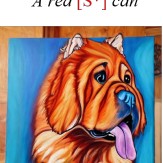

*dog*

A [S*] dog wearing a black top hat and a monocle

A tech-savvy [S*] dog on the office desk

A [S*] dog in a an astronaut outfit

A vibrant Van Gogh-inspired [S*] dog painting

Cubist-style portrait of a [S*] dog by Picasso

Georgia O'Keeffe style [S*] dog painting

**Figure 1: Image customization results of our method, which enables the recreation of a specified conceptual subject under unstylized text guidance, while also providing the flexibility to dynamically edit the subject with challenging stylized prompt.**

## ABSTRACT

Image customization involves learning the subject from provided concept images and generating it within textual contexts, typically yielding alterations of attributes such as style or background. Prevailing methods primarily rely on fine-tuning technique, wherein a unified latent embedding is employed to characterize various concept attributes. However, the attribute entanglement renders customized result challenging to mitigate the influence of subject-irrelevant attributes (e.g., style and background). To overcome these issues, we propose Equilibrated Diffusion, an innovative method that achieves equilibrated image customization by decoupling entangled concept attributes from a frequency-aware perspective, thus harmonizing textual and visual consistency. Unlike conventional approaches that employ a shared latent embedding and tuning process to learn concept, our Equilibrated Diffusion draws inspiration from the correlation between high- and low-frequency components with image style and content, decomposing concept accordingly in the frequency domain. Through independently optimizing concept embeddings in the frequency domain, the denoising model not only enriches its comprehension of style attribute irrelevant to subject identity but also inherently augments its aptitude for accommodating novel stylized descriptions. Furthermore, by combining different frequency embeddings, our model retains the spatially original customization capability. We further design a diffusion process guided by subject masks to alleviate the influence of background attribute, thereby strengthening text alignment. To ensure subject-related information consistency, Residual Reference Attention (RRA) is incorporated into the denoising model of spatial attention computation, effectively preserving structural details. Experimental results demonstrate that Equilibrated Diffusion surpasses other competitors with better subject consistency while closely adhering to text descriptions, thus validating the superiority of our approach. The code is available at https://github.com/MAPLE-AIGC/EqDiff.

*Guo-Jun Qi is the corresponding author.

## CCS CONCEPTS

• **Computing methodologies → Computer vision**.

## KEYWORDS

Image Customization, Diffusion Model, Fourier Transformation

**ACM Reference Format:**
Liyuan Ma, Xueji Fang, and Guo-Jun Qi. 2024. Equilibrated Diffusion: Frequency-aware Textual Embedding for Equilibrated Image Customization. In *Proceedings of the 32nd ACM International Conference on Multimedia (MM '24), October 28-November 1, 2024, Melbourne, VIC, Australia.* ACM, New York, NY, USA, 9 pages. https://doi.org/10.1145/3664647.3680729

## 1 INTRODUCTION

Text-to-Image (T2I) customization aims to customize a Text-to-Image diffusion model using a handful of provided concept images to generate diverse images aligned with the target prompts. This technology can support a variety of downstream applications, spanning from virtual photography to personalized e-commerce product design. It is noteworthy that the primary challenge in this task lies in striking a balance between maintaining consistency in the subject identity and coherence with the target prompts.

Current T2I customization paradigm mainly encompasses two main approaches: model finetuning and encoder-based pretraining. Encoder-based approaches [12, 43, 47] usually leveraged large-scale datasets tailored to specific domains to learn concepts. However, these methods face challenges in generalization across diverse domains and necessitate substantial image datasets, thereby compromising their adaptability. In contrast, finetuning-based methods [1, 11, 20, 24, 32, 42] established associations between concept and latent textual embedding through denoising optimization. However, textual representations are constrained to overfit given concept and lead to entanglement with identity-unrelated style and background attributes, which undermines its capacity to accommodate new textual descriptions during customization. Although some works [3, 6] aim to mitigate the interference of identity-related information by leveraging distinct embeddings. They primarily focus on spatial attributes such as pose and are unable to address the influence of style attribute. Furthermore, textual representations lack spatial expressiveness, resulting in poor image alignment. We attribute the challenge of achieving the trade-off between image- and stylized prompt-alignment in customization to the interplay between content and style during the optimization. This interplay necessitates the fitting of model parameters representing specific concepts to both content and style, making it difficult to prevent identity-related attribute from being influenced by the style attribute of original concept image.

In this paper, we propose Equilibrated Diffusion to address aforementioned issues, which succeed in image- and text-aligned customization as shown in fig. 1. It incorporates a comprehensive training strategy comprising Frequency-aware Decoupled Textual Embedding (FDTE) and a Mask Guided Diffusion Process (MGDP) to enhance text-alignment by disentanglement between identity-irrelevant attributes (e.g., style and background) with identity-relevant one, along with the Residual Reference Attention (RRA) to enhance image alignment. Motivated by the fact that image style and content can be represented by high- and low-frequency components [15], FDTE decouples the concept into different textual embeddings, each bound to the denoising process of different frequency bands of the input image. Through the independent optimization of content and style embeddings and denoising learning applied to image inputs across various frequency bands, the denoising model not only enhances its understanding of style attributes

unrelated to subject identity but also intrinsically augments its ability to accommodate novel stylized descriptions as evidenced by results presented in fig. 8. Furthermore, MGDP utilizes the mask of subject to restrict concept denoising learning in the subject region, allowing the model to focus on learning the subject's concept and eliminate background interference. Finally, to improve the spatial details preservation of subject concept, our proposed RRA employs a spatial attention mechanism to explicitly inject spatial structural details from encoded reference into denoising features.

In summary, our approach aims to achieve subject consistency and alignment with textual descriptions in the customization of image concepts, offering the following contributions. Firstly, we propose FDTE and MGDP in our training strategy to mitigate interference from irrelevant attributes such as style and background on concepts. These techniques aid the model in learning identity-related concepts while excluding interference from style and background attributes irrelevant to identity. Secondly, we design RRA to preserve texture details of the subject concept in generated images with spatial attention mechanisms. Finally, we demonstrate that Equilibrated Diffusion outperforms prior methods in achieving consistency between identity and prompt, showing qualitative and quantitative advantages.

## 2 RELATED WORK

### 2.1 Diffusion-based Text-to-Image Generation

Diffusion models [16, 31, 36, 38] constitute a category of generative models that learn to simulate the generation process through sequential denoising process, which restores the noisy data disturbed by forward diffusion. Significant advancements have been achieved in image generation based on diffusion models. Subsequent endeavors have focused on enhancing the controllability of generated outputs by incorporating additional conditional information [41, 44] (such as semantic map, sketch, and text, etc.) to govern the generation process. Among these, text conditioning inherently offers superior editability and flexibility. Concurrently, with advancements in large language models, the understanding of text has also improved, thereby fostering the development of text-to-image generation models [25, 29, 33]. Nevertheless, despite the potential for diverse image generation guided by text, text-conditioned image generation models encounter difficulties in tailored image synthesis related to specific concepts. Customizing images entails preserving the identity information correlated with the provided textual descriptions, posing a significant challenge for text-to-image generation models.

### 2.2 Image Customization

Leveraging the power of text-to-image diffusion model, image customization [1–3, 6, 10–13, 20, 24, 32, 42] has witnessed rapid development in controllability and faithfulness. Given a set of concept images, image customization is designed to produce new images of a given subject that adhere to textual descriptions. The core of image customization largely revolves around the adjustment of parameters within the foundational text-to-image generation model or incorporating additional parameters. Such modifications involves fine-tuning text embedding [1, 11, 42], adjusting parameters within the full or partial U-Net network [8, 23, 32, 39], as well

as integrating extra encoder components [7, 9, 14, 18, 34, 43, 45, 47]. Early methods [1, 10, 11, 13, 42] capture the concept of the subject through text embeddings represented by specific token. Typically, Textual Inversion [11] encapsulates the concept with optimized token embedding and generates the customized concept image by composing the optimized token with target prompt. Dream-Booth [32] and Custom diffusion [20] has opted for fine-tuning the U-Net parameters, which results in improved subject fidelity. Another line of work has focused on exploring fast customization generation by additional encoder which requires substantial data and usually focuses on close-domain datasets such as human face. Our work mainly involves general customization and thus is compared with open-domain method Elite [45] in the experiment. In response to the limited expressiveness of spatial structures in text embeddings, several methods [4, 17, 24, 37] have explored injecting concept information at the self-attention layer. However, the generated results tend to entirely retain the reference subject, leading to the loss of text alignment. Recently, some works [3, 6] propose to distinguish identity-irrelevant attributes from identity-relevant ones. While they mainly focus on the impact of pose and background on the subject, our method prioritizes stylistic attributes.

## 3 PRELIMINARY

*Diffusion Models.* Our method is based on the pretrained Stable Diffusion (SD) [31], which is a popular text-to-image model and comprises two main components. Initially, an autoencoder with encoder $\mathcal{E}(\cdot)$ and decoder $\mathcal{D}(\cdot)$ learns to compress input image $I$ into a lower-dimensional latent space $z_0 = \mathcal{E}(I)$, which is then decoded back into image $\mathcal{D}(\mathcal{E}(I)) \approx I$. Subsequently, a conditional diffusion model $\epsilon_\theta$ is trained on this latent space to denoise noisy latent codes conditioned on textual input $y$. The training process involves employing a basic mean-squared loss, denoted as:

$$\mathcal{L}_{LDM} = \mathbb{E}_{z_0 \sim \mathcal{E}(I), y, \epsilon, t} \left[ \| \epsilon - \epsilon_\theta \left( z_t, t, y \right) \|_2^2 \right], \tag{1}$$

where $\epsilon \sim \mathcal{N}(0, I)$ signifies unscaled noise, $z_t$ denotes the latent state at time step $t$. During inference, Gaussian noise $z_T$ is progressively diminished to $z_0$ and then decoded back into image output.

Conditional diffusion model $\epsilon_\theta$ of SD is based on a U-Net architecture, which comprises convolution layers, cross attention layers and self attention layers [40]. Among them, the self attention layers capture the spatial relationship of image features within themselves. Formally, after projecting the latent features in timestep $t$ into queries $q_t$, keys $k_t$ and values $v_t$, the self attention calculation is expressed as follows:

$$\text{SA}(q_t, k_t, v_t) = \text{SoftMax}(\frac{q_t k_t{}^T}{\sqrt{d}}) v_t, \tag{2}$$

where $d$ is the feature dimension of $q_t$, $k_t$, and $v_t$.

*Image Customization.* Image customization [1, 11, 12, 20, 24, 32, 43] learns a trainable text embedding $S^*$ to represent the specific concept based on user-provided concept images. Leveraging $S^*$, novel images embodying the desired concept can be synthesized under diverse textual descriptions. Previous methods [20, 24, 32] further enhance concept representation during optimization by

updating weights of projection matrix of key and value in the cross-attention layer or the entire U-Net's parameters. To effectively integrate $S^*$ into the generative process, the input text prompt $y$ is formulated as 'Photo of a $S^*$ [class]', where [class] corresponds to the specific class name of the designated concept.

*Image Fourier Transformation.* The Fourier transform of images constitutes a fundamental technique widely used in image processing and computer vision fields [19, 22, 27, 35]. It serves as a powerful tool for analyzing the frequency components of an image and extracting important features for various applications. For an image $I \in R^{H \times W \times C}$, the Fourier transform is defined as:

$$\mathcal{F}(I)(u, v) = \frac{1}{\sqrt{HW}} \sum_{h=0}^{H-1} \sum_{w=0}^{W-1} I(h, w) e^{-j2\pi \left( \frac{h}{H} u + \frac{w}{W} v \right)}, \tag{3}$$

where $\mathcal{F}$ denotes the frequency-domain representation of the image and $\mathcal{F}^{-1}$ signifies the corresponding inverse Fourier transformation. High-pass and low-pass filters are conventionally applied to the frequency-domain representation of images to isolate specific frequency bands. The high-pass filter, denoted as $H(u, v)$, serves to suppress low-frequency components while retaining high-frequency details. Conversely, the low-pass filter $L(u, v)$ is designed to preserve low-frequency information while mitigating high-frequency noise. These filters are formulated as follows:

$$H(u, v) = \begin{cases} 0 & \text{if } \sqrt{(u - u_0)^2 + (v - v_0)^2} \leq f_c \\ 1 & \text{if } \sqrt{(u - u_0)^2 + (v - v_0)^2} > f_c \end{cases}, \tag{4}$$
$$L(u, v) = 1 - H(u, v),$$

where $(u_0, v_0)$ represents the center of the Fourier frequency map and $f_c$ is the cutoff frequency. In our work, we utilize Fourier transformation to acquire conceptual images characterized by high and low frequencies, subsequently leveraging them to enhance the model's denoising capacity across various frequency spectra.

## 4 METHODOLOGY

### 4.1 Frequency-aware Decoupled Textual Embedding

Essentially, conventional methods for customizing images through fine-tuning rely on a single text embedding and diffusion learning process applied to the original image to capture image concept. However, these approaches struggle to adapt the concept representation derived from text embedding to customize images under significantly different stylized textual descriptions from the original image. We attribute this challenge to the model's failure to explicitly decouple image content from semantic style during the concept learning process, resulting in a tightly coupled representation. Consequently, we propose Frequency-aware Decoupled Textual Embedding to address this issue. FDTE decouples high-frequency style and low-frequency content information from a frequency-domain perspective, thereby learning and reinforcing separate conceptual representations of content and style through corresponding diffusion learning processes on different frequency bands. As shown in fig. 2, FDTE first transforms the original image $I_{ori}^g$ into high-frequency and low-frequency components $I_{HF}^g$ and $I_{LF}^g$ through image Fourier transformation, accompanied by two

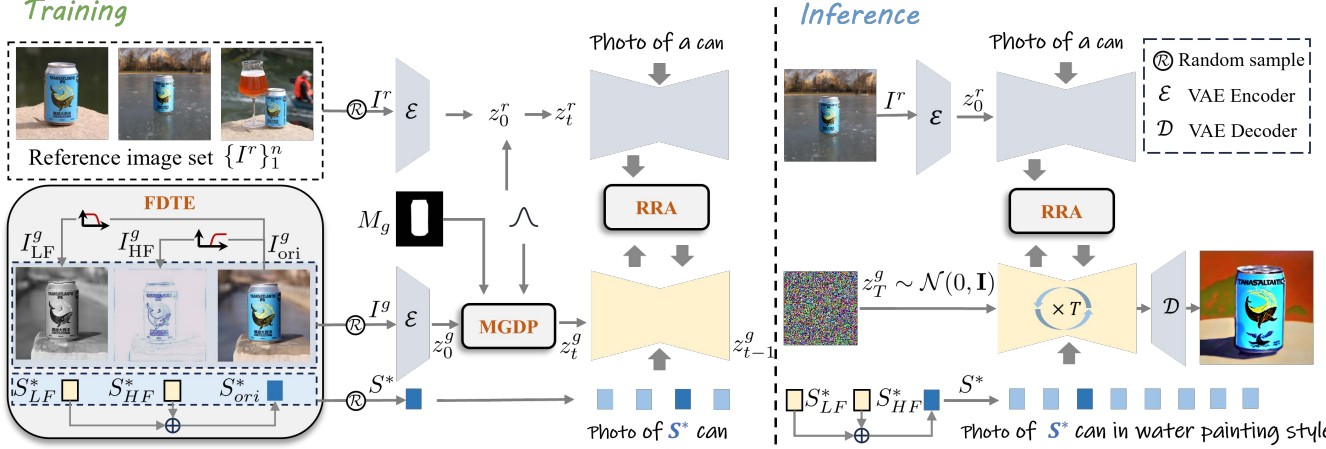

**Figure 2: Method overview. In the training stage, the Frequency-aware Decoupled TextualEmbedding (FDTE) decomposes the original image $I^g_{ori}$ into low frequency (LF) and high frequency (HF) bands, which are selectively formed into the input of denoising UNet along with their corresponding text embeddings including $S^*_{LF}$ and $S^*_{HF}$. After encoding into the latent space, the target features undergo forward diffusion added with noise in Mask Guided Diffusion Process (MGDP), thereby mitigating the impact of background elements. Subsequently, Residual Reference Attention (RRA) integrates spatial details from the reference image $I^r$ into the result to enhance image consistency. During inference, we first generate corresponding spatial domain embedding $S^*$ by optimized high-frequency and low-frequency text embeddings. Then, the sampled Gaussian noise $z^g_T$ is iteratively denoised into generation result aligned with the text for $T$ steps.**

learnable text embeddings $S^*_{HF}$ and $S^*_{LF}$ to indicate respective denoising conditions. Specifically, $S^*_{HF}$ encapsulates considerations regarding high-frequency aware style attribute, while $S^*_{LF}$ attends to low-frequency content attribute. Additionally, to maintain denoising efficacy for the original image $I^g_{ori}$, we retain it as a candidate input for the model, represented by textual embeddings $S^*_{ori}$ that calculated by the sum of frequency-aware embeddings. Formally, the input candidates of denoising model are denoted as follows:

$$I_g = \mathcal{R}_{p_l,p_h,p_o}[L(\mathcal{F}(I^g_{ori})), H(\mathcal{F}(I^g_{ori})), I^g_{ori}],$$
$$S^* = \mathcal{R}_{p_l,p_h,p_o}[S^*_{LF}, S^*_{HF}, S^*_{ori}], where S^*_{ori} = S^*_{LF} + S^*_{HF}, \quad (5)$$

where the operation symbolized by $\mathcal{R}_{p_l,p_h,p_o}[\cdot,\cdot,\cdot]$ entails the random selection of one of the three candidates based on probabilities $p_l$, $p_h$, $p_o$ to serve as the output. During the training phase, these inputs are randomly selected to constitute $I_g$, from which the latent encoding $z^g_0$ is derived. Subsequently, their corresponding text embeddings, which respectively represent high-frequency perception, low-frequency perception, and the original image, are integrated into the fine-tuning process. Notably, during the inference stage, the text embedding $S^*_{ori}$ corresponding to the original image serves as a conditional input to ensure that the generated outcomes adhere closely to real images.

## 4.2 Mask Guided Diffusion Process

While FDTE aids in diminishing the model's reliance on the original image style, direct learning of conceptual representations during the image diffusion process may be interfered by subject-irrelevant

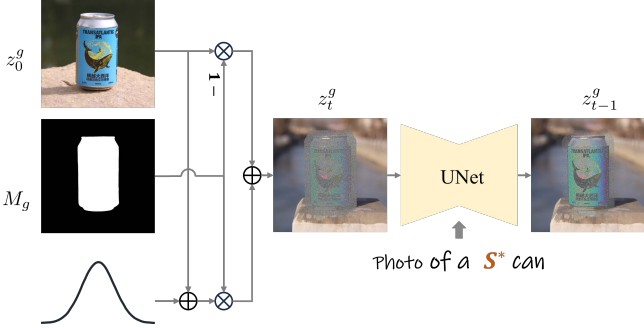

**Figure 3: Illustration of Mask Guided Diffusion Process. To eliminate background interference on the concept representation $S^*$, we exclusively apply noise addition and prediction to the subject region indicated by subject mask $M_g$.**

background attribute. Although previous works [2, 20] have explored strategies wherein the computation of diffusion loss is confined exclusively within the subject region guided by accurate mask, the generated results are not promising as shown in fig. 8. Inspired by SmartBrush [46], we propose a mask-guided diffusion process guided by the subject mask $M_g$, thus compelling the model to focus on learning the conceptual representation of the subject itself. Specifically, as shown in fig. 3, we conduct noise addition and prediction within the subject mask region in the diffusion process and retain clean background information inputted into UNet, which prompts the learning of subject concept. Mathematically, the input

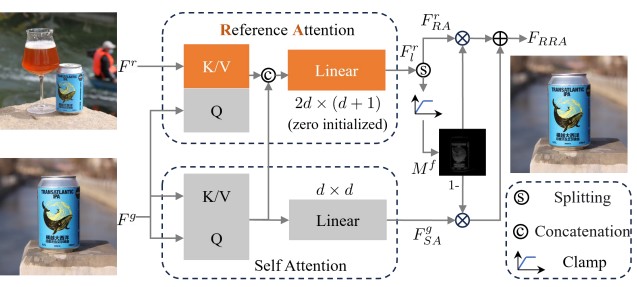

**Figure 4: Framework of Residual Reference Attention (RRA).**

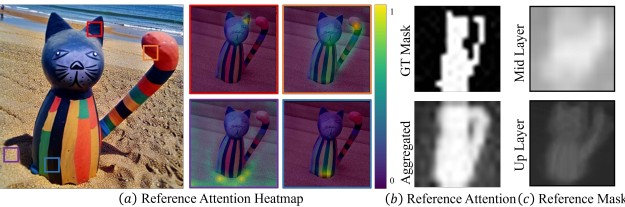

(a) Reference Attention Heatmap          (b) Reference Attention (c) Reference Mask
                                              Loss Analysis

**Figure 5: (a) Reference Attention is capable of attending to reasonable texture information from reference features. (b) The aggregated reference attention map closely reassembles the ground truth subject mask during $\mathcal{L}_{ra}$ calculation eq. (10). (c) Reference Mask $M_f$ predicted in RRA accurately distinguishes the subject from the background.**

noisy latent of UNet is formulated as follows:

$$z_t' = M_g * z_t + (1 - M_g) * z_0, \tag{6}$$

Finally, we reformulate the denoising objective to be aligned with masked diffusion loss [2] and express it as follows:

$$\mathcal{L}_{Mask-LDM} = \mathbb{E}_{z_0 \sim \mathcal{E}(I), y, \epsilon, t} \left[ M_g * \left\| \epsilon - \epsilon_\theta \left( z_t', t, y \right) \right\|_2^2 \right]. \tag{7}$$

### 4.3 Residual Reference Attention

As illustrated in fig. 4, we propose Residual Reference Attention to enhance image alignment by incorporating spatial details of the subject from reference image with reference attention. In the training process of the model, the features of the reference image $I^r$ are initially encoded to obtain the latent code $z_0^r$, which is then perturbed with Gaussian noise of $t$ step to obtain $z_t^r$. This noisy latent code is then fed into a frozen SD UNet $\epsilon_r$. Before the self-attention module of $\epsilon_r$, reference features are extracted and merged with the corresponding features at the positions of UNet $\epsilon_\theta$ containing trainable parameters through RRA, ensuring the proximity and effectiveness of feature fusion in the feature space. During the inference phase, features from the reference image are injected into the denoising network, gradually incorporating spatial information from the reference image during the denoising process over $T$ steps.

In typical personalized tasks, occlusions and variations in viewpoint often occur between reference images and generated results. Consequently, simple fusion of reference image features may introduce interference. Previous approaches [17, 24] only focused on the foreground regions of the subject, neglecting the influence

of backgrounds and mismatched foregrounds, or requiring additional forward processes [4] during inference to obtain the subject's mask for fusion. To address this issue and mitigate additional inference overhead, RRA learns explicit fusion masks and adaptively determines fusion coefficients during the optimization process. As illustrated in Figure fig. 4, RRA projects reference appearance features $F^r \in \mathbb{R}^{hw \times d}$ into keys and values, sharing the same queries from target features $F^g \in \mathbb{R}^{hw \times d}$ within the corresponding self-attention modules. Subsequently, the attention computation results from both reference attention and self attention modules are concatenated as inputs for reference mask $M^f$ prediction and feature calculation, which is formulated as follows:

$$F_l^r = \mathcal{H}_r([\mathrm{SA}(F^g, F^r, F^r), \mathrm{SA}(F^g, F^g, F^g)])$$
$$F_{RA}^r, M^f = \langle F_l^r \rangle_{0:d}, \phi(\langle F_l^r \rangle_{d:d+1}) \tag{8}$$
$$F_{SA}^g = \mathcal{H}_s(\mathrm{SA}(F^g, F^g, F^g)),$$

where $\langle \cdot \rangle$ and $[\cdot]$ mean the feature splitting and concatenation operations, respectively. $\phi(\cdot)$ denotes the clamp function, constraining the results to the interval $(0, 1)$ for generating reference mask $M^f$ as visualized in fig. 5 (c). Reference attention mask $M^f \in \mathbb{R}^{hw \times 1}$ and reference feature $F_{RA}^r \in \mathbb{R}^{hw \times d}$ are calculated by linear layer $\mathcal{H}_r : \mathbb{R}^{2d} \to \mathbb{R}^{d+1}$, which is trainable and zero initialized to promote the training stability at the beginning. The target feature $F_{SA}^g$ is acquired by frozen linear layer $\mathcal{H}_s : \mathbb{R}^d \to \mathbb{R}^d$. The reference attention output can be regarded as the residual information for appearance reference, which is adaptively fused to make the best use of accessible information and is conducted as follows:

$$F_{RRA} = F_{SA}^g * (1 - M^f) + F_{RA}^r * M^f. \tag{9}$$

*Reference Attention Loss.* The preservation of subject details is built on the precise correspondence learned by reference attention. To facilitate the target feature attend more relevant area in reference feature map, we introduce the Reference Attention Loss that encourages the target features within the subject area to have a high attention score distributed in the corresponding subject area of paired reference. The detailed formulation of the loss function is denoted as follows:

$$\mathcal{L}_{ra} = \left\| \sum_{j=1}^{hw} \mathcal{A}_{ij} \odot M^r - M^g \right\|_2$$
$$= \left\| \sum_{j=1}^{hw} \left\{ \mathrm{SoftMax}\left( \frac{F^g \cdot F^{rT}}{\sqrt{d}} \right) \right\}_{ij} \odot M^r - M^g \right\|_2, \tag{10}$$

where reference attention map $\mathcal{A} \in \mathbb{R}^{hw \times hw}$ is calculated by the target feature $F^g$ and reference feature $F^r$. $M^r \in \mathbb{R}^{hw \times 1}$ and $M^g$ denote the subject foreground mask of reference and target image extracted by Grounded-SAM [30], respectively. $\odot$ represents the Hadamard product. As depicted in Figure fig. 5 (b), the upper subfigure illustrates $M^g$, whereas the lower sub-figure showcases the aggregated reference attention scores during the training process, denoted as the first term of $\mathcal{L}_{ra}$. Notably, the visual result of them closely aligns, thereby underscoring the effectiveness of reference attention loss. This affirms its capability to facilitate reference attention in better leveraging the subject information present in the reference image, consequently fostering identity preservation.

## 5 EXPERIMENTS

### 5.1 Experimental Settings

*Datasets.* Our experiment is conducted at DreamBooth dataset [32], which comprises 30 subjects spanning various categories, including both non-live and live subjects. In addition, we also include an commonly used case called "cat toy," which has been utilized in many previous works [8, 11, 24, 45]. Each subject consists of 4 to 6 images paired with 55 unstylized prompts from [24] and another 22 stylized prompts generated by LLM [26] for customization. A complete prompt list is provided in supplementary material.

*Metrics.* Our evaluation follows the same setup and calculation procedures as CustomDiffusion [20], which employs three common metrics to assess the effectiveness of models in aligning text and images: CLIP-T, CLIP-I, and DINO-I. CLIP-T measures text-alignment by the feature similarity between the CLIP [28] visual representation of the generated image and the CLIP textual representation of the corresponding prompt. For the image-alignment assessment, CLIP-I and DINO-I are used to evaluate the visual similarity between generated image and concept image using the features extracted from pretrained CLIP and DINO [5] models.

*Implementation Details.* We follow previous method [32] to employ Stable Diffusion v1.4 as the text-to-image model for fair comparison. Our model is trained by an A100 GPU with the AdamW optimizer [21] for 600 steps and the learning rate is set to 1e-5. Data augmentation techniques are applied to reference images to enhance the disparity with the denoising image, including image cropping, rotation, and flipping, etc. During the inference phase, the images are generated using 50 steps of DDIM [36] with classifier-free guidance scale set to 12.5.

### 5.2 Qualitative Evaluation

The qualitative comparison results with prior methods are presented in fig. 6. As can be seen, our method achieves superior customization performance in stylized prompt alignment and subject identity preservation. For textual stylization alignment, our approach excels in handling various challenging stylistic descriptions compared to existing methods. In cases where the style description closely resembles the original image, our method adeptly preserves the local details of the original image while undergoing stylization (e.g, the can in the 1st row of fig. 6), thereby enhancing its identity consistency throughout the stylization process. In contrast, for more unrealistic stylistic descriptions, our method adeptly maintains crucial semantic elements of the subject while enabling flexible and intricate style editing, a tradeoff that remains elusive to other methodologies. For instance, unlike other approaches, our method successfully disentangles from the influence of the original image's style (see the 3rd, 5th, and 7th rows of fig. 6), thereby avoiding the tendency to generate outcomes resembling the original colors and style, thus demonstrating adaptability to novel stylistic descriptions. Regarding the preservation of subject identity, our method leverages the proposed RRA technique, effectively retaining the crucial texture structures of the concept. More qualitative results can be found in supplementary material.

| Method | CLIP-T (↑) | CLIP-I (↑) | DINO-I (↑) |
|---|---|---|---|
| Textual Inversion (TI) [11] | 0.725 | 0.765 | 0.537 |
| | 0.742 | 0.698 | 0.418 |
| DreamBooth [32] | 0.777 | 0.788 | 0.616 |
| | **0.795** | 0.696 | 0.467 |
| CustomDiffusion (CD) [20] | 0.770 | 0.792 | 0.634 |
| | 0.774 | 0.718 | 0.505 |
| ELITE [45] | 0.756 | 0.777 | 0.589 |
| | 0.718 | 0.741 | 0.548 |
| DreamMatcher [24] | 0.768 | 0.792 | 0.637 |
| | 0.766 | 0.703 | 0.503 |
| Ours | **0.782** | **0.810** | **0.670** |
| | 0.790 | **0.755** | **0.584** |

Table 1: Quantitative comparison under $\boxed{\textit{unstylized}}$ and *stylized* prompts. The best and the second best results are bold-faced and underlined.

### 5.3 Quantitative Evaluation

Following this, we conduct a comprehensive evaluation of our method from a quantitative perspective to validate its efficacy. The baseline method for our approach is the custom diffusion. As depicted in table 1, we achieved the highest CLIP-I and DINO-I scores with both stylized and unstylized textual descriptions. This indicates that our method excels in preserving texture information of the subject and maintains superiority in retaining subject identity.

Furthermore, our method demonstrates close-to-leading results in stylized scenarios compared to Dreambooth, while exhibiting optimal CLIP-T score in non-stylized prompts, showcasing its ability for text alignment. This showcases our method's capability for superior textual editability. In summary, our method's leading performance across these metrics attests to its effectiveness in preserving subject identity and aligning with textual descriptions.

### 5.4 Ablation Study

In this section, we conduct ablation study on the core components of our method to verify their contributions. Visual comparison results of various ablation models are presented in fig. 8, while numerical analyses are provided in fig. 7. The evaluation settings of each experiment remain the same as in section 5.1.

*Effect of Residual Reference Attention and $\mathcal{L}_{ra}$.* Compared to the baseline, the visual results in fig. 8 indicate that the variant model B better preserves the main structural details in conceptual images after incorporating RRA, such as the patterns on jars or the appearance of cats. Additionally, improvements are observed in metrics like CLIP-I and DINO-I, which indicate enhanced subject consistency, highlighting the significant efficacy of the proposed RRA. Analysis of the reference attention heatmaps further reveals that the generated results accurately attend to corresponding positions in the reference images, validating the effectiveness of the RRA.

Moreover, reference attention loss $\mathcal{L}_{ra}$ further improves image-alignment metrics. Visual analysis of the aggregated reference attention maps demonstrates that the subject area can effectively focus on the corresponding areas in the reference, as evidenced by the comparison between the aggregated reference attention map

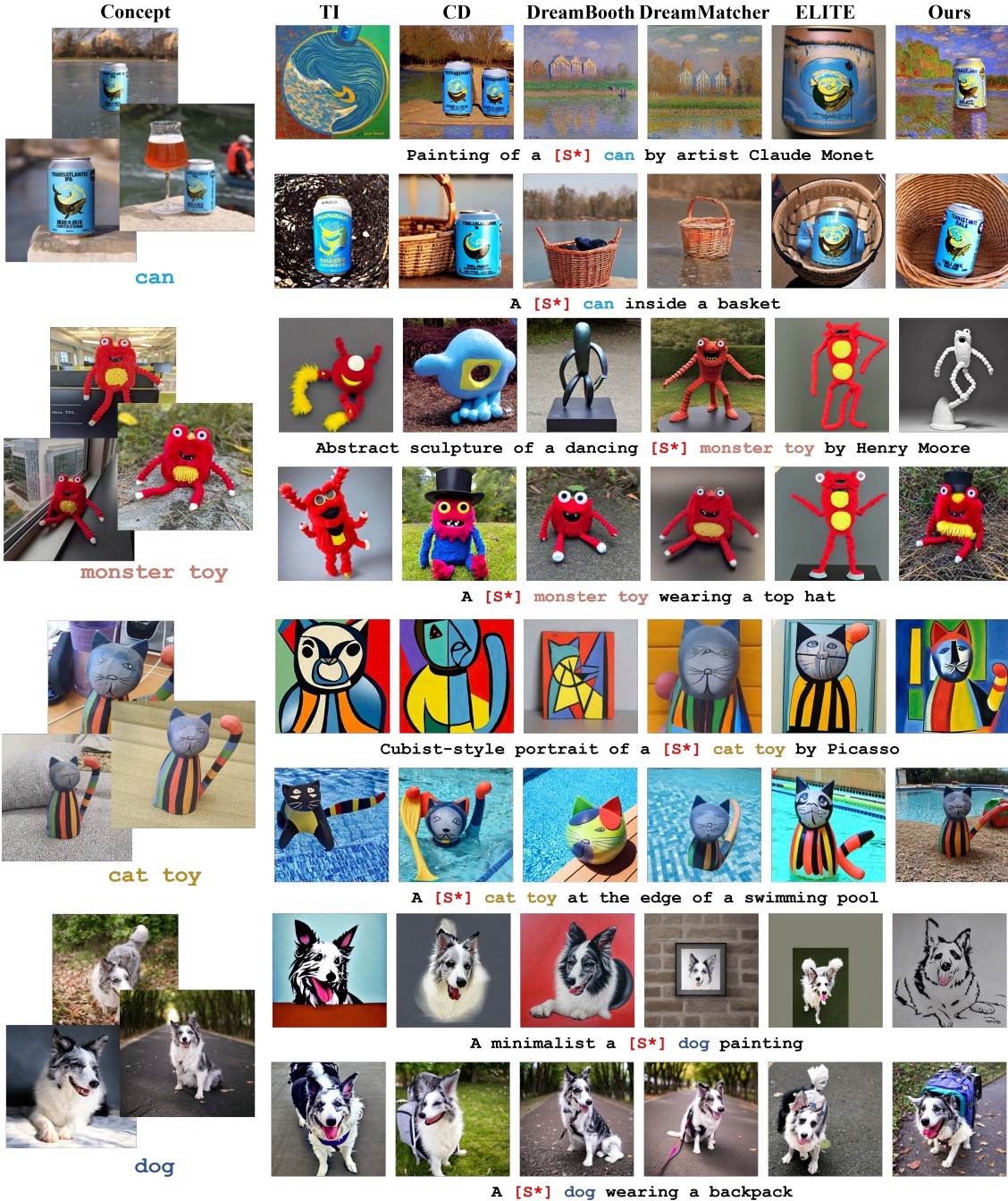

**Figure 6: Qualitative comparison with prevalent methods. Please zoom in for a better view.**

and the ground truth attention map in fig. 5 (B), thus proving the effectiveness of reference attention loss.

*Effect of Frequency-aware Decoupled Textual Embedding.* Subsequently, we conduct an analysis of FDTE's impact on the effectiveness of stylized descriptions. As depicted in fig. 8, the absence of FDTE results in generated outputs that are still entrenched in the original conceptual style, thereby struggling to conform to new stylized textual descriptions (as evidenced by the cat in fig. 8 maintaining its original image style). Conversely, FDTE facilitates the expression of stylistic effects. Additionally, FDTE also leads to a discernible enhancement in the CLIP-T score, as shown in fig. 7. This affirms FDTE's capability in bolstering textual coherence.

| | Ablation Models | CLIP-T ($\uparrow$) | CLIP-I ($\uparrow$) | DINO-I ($\uparrow$) |
|---|---|---|---|---|
| A | Baseline (CD [20]) | 0.770 | 0.792 | 0.634 |
| | | 0.774 | 0.718 | 0.505 |
| B | A + RRA | 0.755 | 0.822 | 0.690 |
| | | 0.746 | 0.755 | 0.591 |
| C | B + $\mathcal{L}_{ra}$ | 0.755 | **0.823** | **0.691** |
| | | 0.749 | **0.757** | **0.595** |
| D | C + FDTE | 0.758 | 0.805 | 0.656 |
| | | 0.772 | 0.717 | 0.513 |
| E | D + MDL [2] | 0.760 | 0.811 | 0.670 |
| | | 0.765 | 0.726 | 0.537 |
| F | D + MGDP (Ours) | **0.782** | 0.810 | 0.670 |
| | | **0.790** | 0.755 | 0.584 |

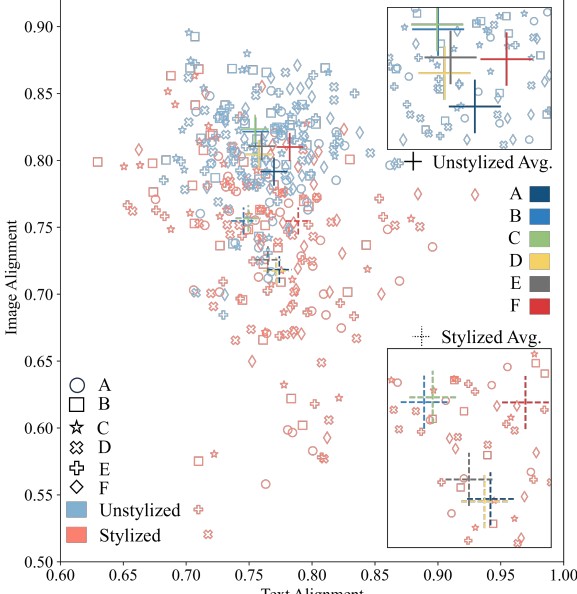

Figure 7: The best and the second best results are bold-faced and underlined. (Upper) Numerical analysis with unstylized and *stylized* prompts. (Lower) Metrics across all classes are visualized, while the averaged metrics are highlighted on the right. Compared to other model variants, our model tends towards the upper-right quadrant, indicative of achieving superior text- and image-alignment trade-offs.

To investigate FDTE's hyperparameter settings in detail, we analyze the probabilities of selecting high-frequency, low-frequency, and original images. For unstylized and stylized scenarios, using $p_l, p_h, p_o = [0.1, 0.1, 0.8]$ achieves superior metrics, indicating an optimal balance between image fidelity and textual coherence.

*Effect of Mask Guided Diffusion Process.* Finally, we analyze the efficacy of MGDP in mitigating the interference of background on the generated results. As illustrated in fig. 8, RRA tends to capture spatial information from concept images directly, thereby rendering the background more similar to the conceptual image (see A and B in fig. 8). Exclusion of MGDP leads to overfitting to the concept

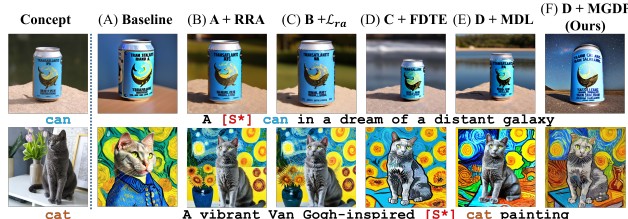

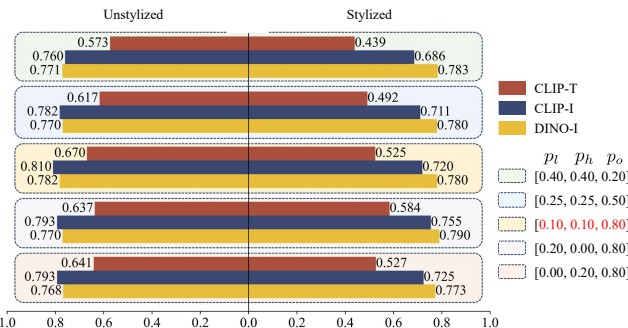

Figure 8: Visualization results of ablation study.

Figure 9: Numerical analysis of FDTE. $p_l$, $p_h$, and $p_o$ represent the probability of choosing low-frequency, high-frequency and original images, respectively.

image's background in the result. Although MDL introduces constraints on background regions in loss objectives, it still struggles to mitigate the influence of background attributes. Conversely, MGDP exhibits superior capability in decoupling backgrounds from the subject, aligning the generated results more closely with textual descriptions (see the galaxy background in the last column of fig. 8).

Notably, our goal is to strike a balance between aligning images and text. Although our full setting shown in the upper table of fig. 7 does not yield optimal scores across all metrics compared to other model variations, the lower figure of fig. 7 reveals that our approach is positioned closer to the upper-right quadrant. This observation demonstrates the superiority of our method in achieving the desired trade-off between image- and text-alignment.

## 6 CONCLUSION

In conclusion, we propose Equilibrated Diffusion to customize images for better image consistency and stylized text alignment. Specifically, by decoupling content and style through frequency-aware decoupled textual embedding, we decompose the original diffusion optimization process across different frequency bands. This enhances the model's ability to understand content represented by low frequencies and style represented by high frequencies, which is guided by decoupled text embeddings and enhances text consistency expression. The mask guided diffusion process mitigates the influence of the concept image background on the results and further enhances text alignment. Moreover, the residual reference attention and reference attention loss better transfer spatial details from reference concepts, promoting texture consistency.

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
