# OpenReview forum: "Equilibrated Diffusion: Frequency-aware Textual Embedding for Equilibrated Image Customization"
_acmmm.org/ACMMM/2024/Conference — MM2024 Oral_

### Official Review · Reviewer_qSuy · 2024-05-19

**Rating:** 5
**Confidence:** 2

**Summary:**

The paper presents Equilibrated Diffusion, a novel architecture tackling the task of image customization. The authors state that the main challenge regards the entanglement between content and style, and previous methods fail to decouple these two aspects. To solve such issues, the authors introduce various components in their architecture. In particular, a frequency-aware decoupled textual embedding separates high-frequency style from low-frequency content, thus reducing the reliance on the original image style and allowing the representation of diverse styles. In addition, mask guidance helps to avoid interference from the background, while Residual Reference Attention (along with a reference attention loss) is applied to incorporate spatial details of the reference image to preserve the subject consistency.
The proposed method is evaluated both qualitatively and quantitatively against state-of-the-art approaches, showing promising results in terms of both subject preservation and alignment with different textual prompts representing diverse styles and contexts. An ablation study assesses the contribution of each architectural component.

**Strengths:**

## Strengths

* The paper is well-presented and structured. I appreciated the good analysis of the current challenges and the problems faced by previous approaches.

* The idea of using frequency-aware Decoupled Textual Embedding and residual reference attention is interesting and well-motivated.

* Overall, the experimental evaluation is well presented, and the model is evaluated considering different aspects: qualitative and quantitative comparison against several SOTA methods, image-text alignment, ablation study to investigate the contribution of each single component. Results are also validated through a user study that corroborates the superiority of the "Equilibrated Diffusion" model.

**Limitations:**

## Major Weaknesses

* I believe that including a section on limitations would benefit the quality of the paper. In this paper, such a section is missing. In particular, I am interested in the possible failure cases of the model. Do the authors have any idea for which cases RRA and  FDTE could fail? I would appreciate a discussion that goes beyond the fact that the model could inherit the classical limitations of Stable Diffusion (e.g. hands, text,...), but instead mainly focuses on the mentioned components of the proposed architecture.

* Although the experimental evaluation is extensive and well conducted, metrics to evaluate the final image quality are missing (e.g. FID [1], KID [2],...)

[1]: GANs Trained by a Two Time-Scale Update Rule Converge to a Local Nash Equilibrium, NeurIPS 2017.

[2]: Demystifyng MMD GANs, ICLR 2018

## Minor Weaknesses

Additional comments to improve the overall quality of the paper (not influencing the final recommendation):

* In Equation 8 (last row), should not the term be $F^g_{SA}$  instead of $F^{r}_{SA}$?

* Typo line 579: ”assessment” instead of ”assessment”.

**Suitability:**

3

---

### Official Review · Reviewer_FaNL · 2024-05-20

**Rating:** 5
**Confidence:** 2

**Summary:**

The presented paper tackles the image customization problem. The aim of this task is to learn a subject from some concept images and generate it within a textual context. Previous methods struggle to disentangle subject-irrelevant attributes from the concept itself. In this setting, the proposed approach leverages the correlation between high- and low-frequency components with image style and content to disentangle the concept information. The presented methods propose a diffusion process guided by subject masks to alleviate the background influence.

**Strengths:**

- ablation experiments, quantitative, and qualitative results support the paper's claims
- the paper presents a user study that confirms the claims
- frequency analysis already proved to be a helpful instrument in the image generation [1] task, therefore supporting the claim


[1] Suvorov, R., Logacheva, E., Mashikhin, A., Remizova, A., Ashukha, A., Silvestrov, A., ... & Lempitsky, V. (2022). Resolution-robust large mask inpainting with Fourier convolutions. In Proceedings of the IEEE/CVF winter conference on applications of computer vision (pp. 2149-2159).

**Limitations:**

- [547] It is not clear what M^r is. There are no references in the figures or in the text. Is it perhaps M^f
- Figure 7 uses abbreviations for the names of the methods (i.e., A, B, C, D), but these abbreviations are not defined in the text. I suppose they are defined in Fig. 8 (a). Nonetheless, I advise you to define them before using them.
- More qualitative examples in the supplementary can help the reader to understand the goodness/limitations of the proposed method

**Suitability:**

3

---

### Official Review · Reviewer_o1Wd · 2024-05-24

**Rating:** 4
**Confidence:** 3

**Summary:**

The paper focus on the issue of the entanglement between the content and style in image customization task. They learn two embeddings in the high and low frequency band respectively to disentangle the content and style. They also propose residual reference attention to enhance image alignment by incorporating spatial details of the subject from reference image.

**Strengths:**

1.The method of learning two embeddings in the low and frequency band respectively is novel and suitable for the entanglement issue.
2.The performance of the method is good in the experiments.

**Limitations:**

There are something that doesn’t make sense in the Residual Reference Attention.
1. Why the attention computation results from both reference attention and self attention modules should be concatenated as inputs for reference mask prediction. It may be enough to predict the reference mask using the reference attention only and there is no ablation study to prove the effectiveness of the concatenation.
2. What’s more, it’s strange to to sum the production of the reference mask and the reference image and the production of the inverse mask and the target image.

**Suitability:**

3

---

### Meta-Review · Area_Chair_cwfm · 2024-07-04

**Recommendation:** Accept (Oral)
**Confidence:** 5

**Metareview:**

The submission addresses the challenging problem of disentangling content and style in image customization tasks. The proposed method introduces innovative solutions such as frequency-aware decoupled textual embeddings and residual reference attention to tackle these challenges effectively. All reviewers have raised valid points, and the authors have satisfactorily addressed most concerns. The method's novelty and the strong experimental results make a compelling case for acceptance. Thus, I recommend a Accept (Poster) for this submission, acknowledging its contributions to multimedia/multimodal processing while suggesting improvements in clarity and comprehensive evaluation in future revisions.